# Retrieval-Retro: Retrieval-based Inorganic Retrosynthesis with Expert Knowledge

**Heewoong Noh**[1], **Namkyeong Lee**[1], **Gyoung S. Na**[2*], **Chanyoung Park**[1*]

[1] KAIST     [2] KRICT

{heewoongnoh,namkyeong96,cy.park}@kaist.ac.kr
ngs0@krict.re.kr

## Abstract

While inorganic retrosynthesis planning is essential in the field of chemical science, the application of machine learning in this area has been notably less explored compared to organic retrosynthesis planning. In this paper, we propose Retrieval-Retro for inorganic retrosynthesis planning, which implicitly extracts the precursor information of reference materials that are retrieved from the knowledge base regarding domain expertise in the field. Specifically, instead of directly employing the precursor information of reference materials, we propose implicitly extracting it with various attention layers, which enables the model to learn novel synthesis recipes more effectively. Moreover, during retrieval, we consider the thermodynamic relationship between target material and precursors, which is essential domain expertise in identifying the most probable precursor set among various options. Extensive experiments demonstrate the superiority of Retrieval-Retro in retrosynthesis planning, especially in discovering novel synthesis recipes, which is crucial for materials discovery. The source code for Retrieval-Retro is available at https://github.com/HeewoongNoh/Retrieval-Retro.

## 1   Introduction

Discovering new materials is a fundamental problem in materials science [28, 4, 25], providing innovative options in various industry fields, such as semiconductors and batteries [37, 36]. On the other hand, it is also important to establish synthetic routes for newly discovered materials [21], i.e., retrosynthesis planning, as the ability to synthesize these materials is essential for their successful commercialization beyond mere discovery.

For organic materials, retrosynthesis planning approaches identify valid and efficient synthetic routes [8] by breaking down complex target molecules into smaller molecules that are commercially available and easily synthesizable. During the process, they focus on the structural information of target molecules, such as functional groups and reaction centers, that are related to widely known organic reaction mechanisms [21, 7, 9, 40, 22, 23]. Following the practice of organic retrosynthesis, machine learning (ML) based approaches utilizing the molecular structure expressed as SMILES strings [43] or molecular graphs have been extensively studied [45, 33].

However, unlike organic retrosynthesis, using the atomic structural information of inorganic materials for retrosynthesis presents significant challenges due to 1) the high computational load incurred by the larger number of atoms compared to organic molecules [7, 6], and 2) the failure of traditional physical theories for the atomic structure computation caused by the inclusion of diverse and unusual elements [11, 1]. Therefore, a chemical composition-based approach is essential for retrosynthesis planning of inorganic materials. Besides the challenges of using the atomic structural information, there is a lack of a clear and general theory regarding the mechanisms of inorganic synthesis reactions

---

[*]Corresponding Author.

38th Conference on Neural Information Processing Systems (NeurIPS 2024).

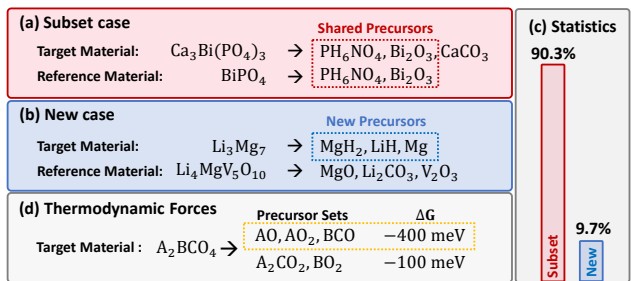

Figure 1: An example case where **(a)** the target material shares a subset of precursors with reference material, and **(b)** the target material has an entirely new set of precursors, without sharing any subset of precursors with reference material. **(c)** The proportion of subset cases and new cases among the materials newly synthesized from 2017 to 2020. **(d)** The target material is more likely to be synthesized using the precursor set that exhibits a more negative driving force.

[34]. For these reasons, inorganic retrosynthesis planning is a more challenging task compared to organic retrosynthesis planning.

Given these challenges inherent in the retrosynthesis planning of inorganic materials, applying existing ML-based organic retrosynthesis methods for inorganic purposes is infeasible. Consequently, there has been limited research into inorganic retrosynthesis planning compared to that for organic materials. As a pioneering work, CVAE [17] generates synthesis variables for target materials using a generative model, and ElemwiseRetro [18] reformulates the precursor prediction task as a multi-class classification problem with dozens of curated precursor templates. However, these approaches overlook the common practices in conventional inorganic material synthesis, where chemists identify reference materials similar to the target material and consult established synthesis recipes [12, 13].

More specifically, due to the aforementioned inherent challenges, chemists often engage in a costly trial-and-error method by referencing precedent recipes of reference materials from prior literature [21, 12]. Therefore, as illustrated in Figure 1 (c), the majority of the discovered synthetic routes for a target material share a common set of precursors with the corresponding reference material from previous studies (Figure 1 (a)), whereas only a small number of these routes involve a completely new set of precursors (Figure 1 (b)). Inspired by the common practice, He et al. [12] propose to retrieve reference materials that are similar to a target inorganic material from a knowledge base of previous studies, and leverage their precursor information for retrosynthesis planning of the target material.

Although the method proposed by He et al. [12] has proven to be effective by emulating standard practices in the field, it faces two significant limitations. The first limitation is that the model's predictive capability is limited to the precursor sets of retrieved reference material, thus inhibiting its capacity to deduce novel synthetic pathway. This is due to the heavy reliance on the precursor sets of reference materials. However, despite a small fraction of materials being synthesized with entirely new synthetic recipes as shown in Figure 1 (c), it is widely known that discovering novel synthetic routes with entirely new precursors can accelerate the inorganic material synthesis process [29, 27] and facilitate the discovery of cost-effective recipes [26, 30]. Thus, despite a large number of materials being synthesized through slight alterations to previously known synthesis recipes due to the complexities of inorganic material synthesis, it remains critical to identify new synthetic pathways that extend beyond the commonly known synthetic recipes.

The second limitation is that it neglects the widely known domain expertise in the field [17, 18, 12]: the greater (more negative) thermodynamic driving force ($\Delta G$) between the target material and the precursor set, the more feasible it is to actually form the target material through the precursor set [31, 35]. As an example in Figure 1 (d), given a target material $A_2BCO_4$, a precursor set $\{AO, AO_2, BCO\}$ exhibits a significantly greater $\Delta G$ compared to another set $\{A_2CO_2, BO\}$, making it more probable that the target material will be synthesized from the first precursor set. Therefore, by analyzing the thermodynamic relationships between the target material and various precursor sets, we can identify which combinations of precursors are most feasible for material synthesis. For example, considering such relationship enables the selection of precursor sets that are likely effective starting materials for synthesizing the target material, thereby facilitating successful synthesis. However, previous works [17, 18, 12] overlook this crucial domain expertise, leading to their inability to identify these optimal precursor sets.

To this end, we propose a novel inorganic retrosynthesis planning approach by implicitly extracting the precursor information with the domain expertise-enhanced reference material retriever. Specifically, instead of directly utilizing precursor information as in He et al. [12]—that is, explicitly incorporating precursors from retrieved materials for prediction—we propose to implicitly extract this information from reference materials using various attention layers that are designed to enhance and extract the precursor details of the reference materials. By providing the model with greater flexibility, we expect it to discover novel synthesis recipes that go beyond existing ones. Moreover, to determine which material should be referenced, we utilize well-established domain expertise in the field, i.e., the thermodynamic relationships between the target material and potential precursors, with a novel Neural Reaction Energy retriever. With a novel retriever, our model effectively identifies which material to refer to for inorganic retrosynthesis planning of the target material. Our extensive experiments demonstrate the effectiveness of Retrieval-Retro in inorganic retrosynthesis planning, especially discovering novel synthetic recipes, demonstrating the potential applicability of Retrieval-Retro in real-world materials discovery.

In this study, we make the following contributions:

- We propose to implicitly integrate the precursor information of reference materials, which enables the model to more effectively discover novel synthetic recipes of inorganic materials.

- Furthermore, we introduce a novel retriever inspired by domain expertise, which assists the model in effectively determining which material to reference during inorganic retrosynthesis planning.

- Extensive experiments demonstrate the effectiveness of Retrieval-Retro in various scenarios, particularly in the year split, which poses a more realistic and challenging environment. Additionally, its exceptional capability in uncovering new synthetic recipes for inorganic materials highlights its potential for practical application in real-world material discovery.

## 2 Preliminaries

### 2.1 Problem setup

**Inorganic Retrosynthesis Planning with Chemical Composition.** In inorganic retrosynthesis planning, determining the atomic structure of inorganic materials poses significant challenges and demands costly computational efforts. As a result, both chemists [21] and prior ML methodologies [18, 17, 12] depend exclusively on composition information. In line with previous studies, we also base our approach on the composition information of inorganic materials instead of structural data.

**Notations.** An inorganic material can be quantitatively described by a composition vector $\mathbf{x} \in \mathbb{R}^d$, where $d$ represents the total number of unique chemical elements. Each element in the vector $\mathbf{x}$ represents the proportion of each chemical element that constitutes the material. As an example, consider the material with chemical formula $SiO_2$, where Si and O correspond to element numbers 14 and 6, respectively. It can be represented as $\mathbf{x} = (x_1, x_2, \ldots, x_d)$, where $x_{14} = \frac{1}{3}$, $x_6 = \frac{2}{3}$, with all other $x$ values being zero. Moreover, following a previous work [10], we construct a fully connected composition graph $\mathcal{G} = (\mathcal{E}, \mathbf{A})$, where $\mathcal{E}$ is the set of elements associated with the nonzero components of $\mathbf{x}$, and $\mathbf{A} \in \{1\}^{n \times n}$ is the fully connected adjacency matrix with $n$ indicating the number of nonzero entries in $\mathbf{x}$. We initialize the initial feature $\mathbf{e}_i$ of each element $e_i$ in $\mathcal{E}$ with Matscholar [44], whose element embedding is obtained from the vast amount of scientific literature.

**Task: Precursor Prediction.** Following a previous work [12], we formulate precursor prediction as a multi-label classification problem. Given a fully connected graph $\mathcal{G} = (\mathcal{E}, \mathbf{A})$ representing an inorganic material, our objective is to train a model $\mathcal{F}$ that predicts the possible precursors for the material, i.e., $\mathbf{y} = \mathcal{F}(\mathcal{G})$, where $\mathbf{y} \in \{0, 1\}^l$ indicates the label vector with each element signifying the predefined $l$ precursors. That is, each element $y_i$ in the label vector $\mathbf{y}$ indicates whether $i$-th precursor is necessary ($y_i = 1$) or not ($y_i = 0$) for synthesizing the target material $\mathcal{G}$.

### 2.2 Composition Graph Encoder

To begin with, we briefly introduce the compositional graph encoder, which is used to encode the material representation in the paper. Specifically, given a composition graph $\mathcal{G} = (\mathcal{E}, \mathbf{A})$ of a material, we obtain the material representation $\mathbf{g}$ as follows:

$$\mathbf{g} = \text{Pooling}(\text{GNN}(\mathcal{E}, \mathbf{A})), \tag{1}$$

where "Pooling" refers to the sum pooling of the node representations within the composition graph $\mathcal{G}$, which are derived from a GNN encoder. The detailed architecture used in the paper is provided in

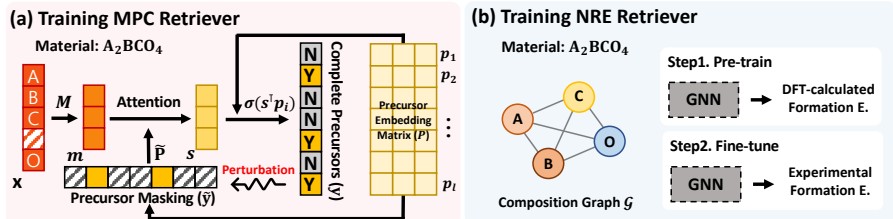

Figure 2: **(a)** Training process of the Masked Precursor Completion (MPC) retriever. **(b)** Training process of the Neural Reaction Energy (NRE) retriever.

Appendix A.1. Moreover, we examine whether the proposed framework can consistently improve in various GNN architectures in Appendix E.1.

## 3   Proposed Method: Retrieval-Retro

In this section, we introduce our proposed method Retrieval-Retro, a novel inorganic retrosynthesis planning approach that implicitly extracts the precursor information of reference materials retrieved with two complementary retrievers, i.e., Masked Precursor Completion (MPC) Retriever and Neural Reaction Energy (NRE) Retriever. The overall framework of Retrieval-Retro is shown in Figure 3.

### 3.1   Reference Material Retrieval

Before we extract precursor information from the reference materials, it is essential to decide which material should be referenced for the extraction. To elaborately determine which materials to reference, we employ two complementary retrievers: the Masked Precursor Completion (MPC) retriever and the Neural Reaction Energy (NRE) retriever.

**Masked Precursor Completion (MPC) Retriever.** MPC retriever identifies the reference materials sharing similar precursors with the target material by learning dependencies between precursors. Specifically, given a chemical composition vector $\mathbf{x}$ of a target material, we obtain its representation $\mathbf{m} = M(\mathbf{x})$, where $M : \mathbb{R}^d \to \mathbb{R}^{d'}$ indicates two layered MLPs with non-linearity. We define a learnable precursor embedding matrix $\mathbf{P} \in \mathbb{R}^{l \times d'}$, whose $i$-th row $p_i \in \mathbb{R}^{d'}$ represents a learnable embedding vector for the $i$-th precursor. Concurrently, we generate a randomly perturbed precursor vector $\tilde{\mathbf{y}}$ from the provided precursor information $\mathbf{y}$, and create a perturbed precursor matrix $\tilde{\mathbf{P}}$ by applying the perturbed precursor vector $\tilde{\mathbf{y}}$ as a mask to the precursor matrix $\mathbf{P}$. Specifically, $\tilde{p}_i$ is masked if $\tilde{y}_i = 0$, and is left unchanged otherwise. We then integrate the representation $\mathbf{m}$ and the perturbed precursor matrix $\tilde{\mathbf{P}}$ with cross-attention to form precursor conditioned representation of the material $\mathbf{s}$. Then, the model is trained to reconstruct the original precursor vector $\mathbf{y}$ from the precursor conditioned representation $\mathbf{s}$ and precursor matrix $\mathbf{P}$, by representing probability for each precursor as follows $\sigma(\mathbf{s}^\top p_i)$. The overall training procedure of MPC retriever is in Figure 1 (a).

By doing so, it enables the retriever to learn dependencies among precursors and the correlation between the precursors and the target material. With the MPC retriever, we calculate the cosine similarity between the representation of the target material and all materials in the knowledge base obtained through $M$, and retrieve the top $K$ materials that are similar to the target material.

**Neural Reaction Energy (NRE) Retriever.** Although the MPC retriever effectively identifies reference materials with potentially similar sets of synthesis precursors, it overlooks widely recognized domain expertise in the field, i.e., the thermodynamic relationships between materials, which is essential for the inorganic synthesis process, particularly in selecting appropriate precursors [35, 27]. More specifically, the thermodynamic driving force between the target material and precursor set can be quantified by Gibbs free energy ($\Delta G$), which is a measure of the material's thermodynamic stability. Under constant pressure and temperature, a negative $\Delta G$ indicates that the energy of the target material is lower than that of the precursor set, signifying that the synthesis reaction can occur spontaneously [31, 35]. As a result, it is widely known that the more negative $\Delta G$, the more precursor set is likely to synthesize the target material.

Based on this knowledge, it would be beneficial to retrieve materials that have the precursor set capable of inducing favorable reactions with the target material by considering the thermodynamic force. $\Delta G$ can be approximated by the difference $\Delta H$ between the enthalpy of the target and

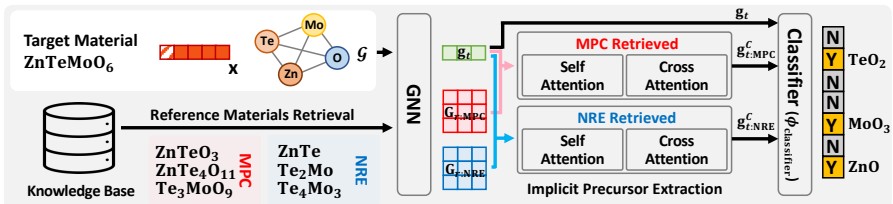

Figure 3: The overall framework of Retrieval-Retro.

precursor set $\Delta H$ as follows:

$$\Delta G \approx \Delta H = H_{Target} - H_{Precursor\ set}, \tag{2}$$

where the $H_{Target}$ and $H_{Precursor\ set}$ represent the formation energy of the target and precursor set. Therefore, a straightforward solution for calculating $\Delta H$ is to utilize the formation energy of the target material and the precursor set that can be directly obtained from the extensive database of structure-based DFT-calculated formation energy. However, it is widely known that DFT-calculated values frequently diverge from experimental data, while actual material synthesis occurs in real-world wet lab settings [16]. Even worse, there is no guarantee that these databases encompass all materials of interest in inorganic retrosynthesis planning. Consequently, it is essential to develop a composition-based formation energy predictor that is specifically designed for experimental data.

To this end, we propose a learnable Neural Reaction Energy (NRE) retriever, which is pre-trained on abundant DFT-calculated formation energy data and then fine-tuned on experimental formation energy data as shown in Figure 1 (b) [16]. Specifically, we initially pre-train the NRE retriever using the Materials Project database [14], training the model to predict DFT-calculated formation energy from representations derived from the composition graph encoder (see Section 2.2). Subsequently, we fine-tune the retriever using experimental formation energy data [32], which allows the model to adapt to experimental data. We demonstrate the effectiveness of the training mechanism in Appendix E.2. Finally, given a trained NRE retriever, we can compute the formation energies of the target material and the precursor set of reference materials in the knowledge base. We then retrieve $K$ reference materials that exhibit the most negative $\Delta G$, selecting from those whose precursors contain the same elements as the target material, along with other common elements such as C, H, O, and N. Note that calculations are performed prior to training, so no additional training costs are incurred.

## 3.2 Implicit Precursor Extraction

Now, we discuss how to extract the precursor information from the reference materials elaborately selected in Section 3.1. While the previous work [12] directly utilizes the precursor information of the reference materials, this limits the model's ability to learn and deduce new synthetic recipes for the target material, which can significantly accelerate the materials discovery and reduce the cost of material synthesis. Therefore, we propose to implicitly extract the precursor information from the retrieved material with various attention layers, i.e., self-attention and cross-attention layers, which aim to enhance the representation of reference materials by considering other reference materials and extract the implicit precursor information from the enhanced representation, respectively.

To do so, we first encode the target material and $K$ reference materials using their associated composition graphs $\mathcal{G}$ via the composition graph encoder introduced in Section 2.2. As a result, we obtain the representation of target material $\mathbf{g}_t \in \mathbb{R}^D$ and the $K$ reference materials $\mathbf{G}_r = [\mathbf{g}_r^1, \ldots, \mathbf{g}_r^K] \in \mathbb{R}^{K \times D}$, where $\mathbf{g}_r^k$ indicates the representation of the $k$-th reference material.

**Reference Enhancing with Self-Attention.** To effectively extract the precursor information from the reference materials, we first enhance the representation of these materials through a self-attention mechanism [39, 24]. This approach encourages the model to selectively determine which information to extract from a particular reference material by considering the relationships among various reference materials. To do so, we first obtain a new matrix for reference materials $\mathbf{G'}_r^0 = [\mathbf{g'}_r^1, \ldots, \mathbf{g'}_r^K]$, where $\mathbf{g'}_r^k = \phi_1(\mathbf{g}_r^k \| \mathbf{g}_t)$ indicates the modified representation of the $k$-th reference material regarding the target material, where $\|$ denotes the concatenation operation, and $\phi_1 : \mathbb{R}^{2D} \to \mathbb{R}^D$ is a learnable MLP. Note that by modifying the representation of the reference material through concatenation with the target material, we allow the model to extract information pertaining to the target material, rather than focusing solely on the reference materials. Then, we implement a self-attention mecha-nism to determine which information to extract from the reference material, taking into account the

relationships among the other reference materials:

$$\mathbf{G'}_r^s = \text{Self-Attention}(\mathbf{Q}_{\mathbf{G'}_r^{s-1}}, \mathbf{K}_{\mathbf{G'}_r^{s-1}}, \mathbf{V}_{\mathbf{G'}_r^{s-1}}) \in \mathbb{R}^{K \times D}, \tag{3}$$

where $s = 1, \dots, S$ indicates the index of the self-attention layers. Different from the conventional self-attention layers [39], we directly utilize $\mathbf{G'}_r^{s-1}$ as query $\mathbf{Q}_{\mathbf{G'}_r^{s-1}}$, key $\mathbf{K}_{\mathbf{G'}_r^{s-1}}$, and value $\mathbf{V}_{\mathbf{G'}_r^{s-1}}$, without any learnable parameters [24]. By analyzing the relationships between the reference materials, the model improves the representations of these materials, thereby supplying more appropriate references to be extracted by the cross-attention layer.

**Reference Selection with Cross-Attention.** Lastly, we extract the implicit precursor information by merging the representation of the target material with that of the enhanced reference materials via a cross-attention mechanism [38, 42]. With cross-attention layers, we expect the model to learn favorable synthesis recipes from reference materials by selectively learning from reference materials with attention weights. More formally, cross-attention layers are formulated as follows:

$$\mathbf{g}_t^c = \text{Cross-Attention}(\mathbf{Q}_{\mathbf{g}_t^{c-1}}, \mathbf{K}_{\mathbf{G'}_r^S}, \mathbf{V}_{\mathbf{G'}_r^S}) \in \mathbb{R}^D, \tag{4}$$

where $c = 1, \dots, C$ indicates the index of the cross-attention layers. Note that we use an enhanced reference material representation $\mathbf{G'}_r^S$ and target material representation $\mathbf{g}_t$ as inputs to the first cross-attention layer, i.e., $\tilde{\mathbf{G}}'^0_r = \mathbf{G'}_r^S$ and $\mathbf{g}_t^0 = \mathbf{g}_t$. Moreover, we also utilize $\mathbf{g}_t^{c-1}$ as query $\mathbf{Q}_{\mathbf{g}_t^{c-1}}$, and the reference material representation $\mathbf{G'}_r^S$ as key $\mathbf{K}_{\mathbf{G'}_r^S}$ and value $\mathbf{V}_{\mathbf{G'}_r^S}$ identical to the self-attention layer, without any learnable parameters. By employing cross-attention layers between the target material and reference materials, rather than the precursor set of the reference materials, the model effectively accesses the synthetic recipes of reference materials without explicitly using precursor information, thus enabling the discovery of novel synthetic recipes for the target material. We employ this implicit precursor extraction process for the reference materials gathered using both the MPC retriever and the NRE retriever, resulting in $\mathbf{g}_{t:\text{MPC}}^C$ and $\mathbf{g}_{t:\text{NRE}}^C$, respectively.

### 3.3 Model Training

Finally, we compute the model prediction $\hat{\mathbf{y}}$ as follows: $\hat{\mathbf{y}} = \phi_{\text{classifier}}(\mathbf{g}_t || \mathbf{g}_{t:\text{MPC}}^C || \mathbf{g}_{t:\text{NRE}}^C)$, where $\phi_{\text{classifier}} : \mathbb{R}^{3D} \to \mathbb{R}^l$ is an MLP with non-linearity. Note that each dimension in $\hat{\mathbf{y}}$, i.e., $\hat{y}_i$, indicates the model's predicted probability of whether precursor $i$ will be included or not. For model training, we adopt Binary Cross Entropy (BCE) loss, which is commonly used for multi-label classification learning [5, 47], as: $\mathcal{L} = -\frac{1}{l} \sum_{i=1}^{l} [y_i \log(\hat{y}_i) + (1 - y_i) \log(1 - \hat{y}_i)]$.

## 4 Experiments

### 4.1 Experimental Setup

**Datasets.** We use 33,343 inorganic material synthesis recipes extracted from 24,304 materials science papers [20] following prior studies [12, 18]. Due to the lack of an extensive database containing inorganic synthesis recipes [20], we use the training set as the knowledge base, following a previous work [12]. Additional details about datasets are provided in the Appendix B.

**Baseline Methods.** We compare Retrieval-Retro with two inorganic retrosynthesis methods (i.e., **He et al. [12]** and ElemwiseRetro [18]), two composition-based representation learning methods (i.e., Roost [10] and CrabNet [41]), and three newly proposed baselines (i.e., Composition MLP and Graph Network [3], Graph Network + MPC) to demonstrate the effectiveness of Retrieval-Retro. The first newly introduced baseline is called **Composition MLP**, which does not retrieve reference materials but instead relies on the composition vector of the material. **He et al. [12]** conducts inorganic retrosynthesis planning by using the MPC retriever to access reference materials based solely on the material's composition vector. **ElemwiseRetro** [18] acquires precursor information through a fully connected graph that represents the constituent elements within the material. Furthermore, two composition-based material representation learning approaches, namely **Roost** [10] and **CrabNet** [41], explore the intricate interactions among elements within materials using message passing and self-attention, respectively. Although these methods are initially designed for property prediction, we have adapted prediction heads so that they can be effectively used for inorganic retrosynthesis planning. We also evaluate two new baselines, **Graph Network** [3] and **Graph Network + MPC**. The former predicts precursors without retrieving reference materials, while the latter does so after retrieving references. As these methods utilize the same backbone GNN structure as in our approach,

Table 1: Overall model performance in (a) Year split and (b) Random split. "Int." denotes whether the model accounts for interactions among constituent elements, while "Retr." indicates whether the model retrieves reference materials. **Bold** indicates the best performance, while underline represents the second best performance.

| Model | Int. | Retr. | (a) Year Split Top-1 | Top-3 | Top-5 | Top-10 | Recall Macro | Micro | (b) Random Split Top-1 | Top-3 | Top-5 | Top-10 | Recall Macro | Micro |
|---|---|---|---|---|---|---|---|---|---|---|---|---|---|---|
| Composition MLP | ✗ | ✗ | 31.60 (1.70) | 34.37 (1.58) | 35.22 (1.43) | 36.56 (0.160) | 31.42 (0.030) | 31.44 (0.060) | 58.56 (0.47) | 62.20 (0.36) | 62.95 (0.029) | 64.32 (0.42) | 54.56 (0.44) | 55.35 (0.57) |
| He et al. [12] | ✗ | ✓ | 45.03 (1.85) | 48.02 (1.86) | 49.11 (1.77) | 51.09 (1.93) | 44.72 (1.83) | 44.75 (1.86) | 61.94 (1.5) | 66.44 (1.48) | 67.46 (1.55) | 68.84 (1.65) | 58.55 (1.45) | 59.35 (1.34) |
| ElemwiseRetro | ✓ | ✗ | 53.45 (0.58) | 57.07 (0.52) | 58.19 (0.72) | 60.84 (0.78) | 53.12 (0.60) | 53.19 (0.60) | 77.23 (0.70) | 80.93 (0.54) | 81.57 (0.67) | 82.78 (0.64) | 72.33 (1.14) | 73.26 (0.99) |
| Roost | ✓ | ✗ | 54.38 (0.75) | 57.82 (0.81) | 58.82 (1.00) | 60.71 (1.15) | 54.01 (0.75) | 54.04 (0.74) | 78.42 (0.91) | 82.32 (0.91) | 83.07 (0.83) | 84.10 (0.66) | 73.38 (1.41) | 74.46 (1.22) |
| CrabNet | ✓ | ✗ | 57.15 (0.77) | 61.60 (0.85) | 62.44 (0.82) | 64.14 (0.86) | 56.79 (0.77) | 56.82 (0.77) | 78.69 | 81.62 | 82.27 | 83.35 | 73.27 (1.21) | 74.28 (0.99) |
| Graph Network | ✓ | ✗ | 58.95 (0.41) | 63.10 (0.63) | 64.07 (0.68) | 66.30 (0.62) | 58.54 (0.42) | 58.61 (0.41) | 77.91 (1.31) | 81.55 (0.98) | 82.37 (0.92) | 83.50 (0.90) | 72.96 (1.53) | 73.88 (1.29) |
| Graph Network + MPC | ✓ | ✓ | 60.01 (1.10) | 64.15 (1.10) | 65.15 (1.17) | 67.19 (0.83) | 59.61 (1.10) | 59.66 (1.10) | 79.09 (1.25) | 82.95 (1.13) | 83.82 (1.19) | 84.97 (0.94) | 73.86 (1.34) | 74.81 (1.23) |
| Retrieval-Retro | ✓ | ✓ | **61.16** (0.38) | **65.92** (0.71) | **67.18** (0.76) | **69.45** (1.03) | **60.97** (0.62) | **61.06** (0.62) | **79.81** (0.68) | **83.62** (0.77) | **84.46** (0.78) | **85.70** (0.88) | **74.61** (0.98) | **75.49** (0.89) |

they can be viewed as ablated versions of Retrieval-Retro. We provide further details about the compared baseline methods in Appendix C.

**Evaluation Protocol.** We perform evaluations under two distinct settings, namely, *random split* and *year split*. Following prior studies, under the random split setting, we randomly split the dataset into train/valid/test of 80/10/10%. On the other hand, under the year split setting [12], the training set includes synthesis recipes from papers published up to 2014, the validation set includes recipes from papers published in 2015 and 2016, and the test set includes recipes from papers published between 2017 and 2020. This setup closely replicates the real-world material discovery conditions, allowing for the evaluation of model performance without the need for costly wet-lab experiments.

Following previous works in retrosynthesis planning [45, 40, 18, 9], we adopt Top-K exact match accuracy to evaluate the effectiveness of Retrieval-Retro. In addition, we also employ Macro and Micro Recall, which are commonly used as metrics in the multi-label classification problem [5, 46]. We provide further details on evaluation protocol in Appendix D.

## 4.2 Empirical Results

**Effectiveness of Retrieval-Retro in Inorganic Retrosynthesis Planning.** In Table 1, we have the following observations: **1)** Modeling the interaction among the constituent elements (**Int.** ✓) proves more effective than simply representing the material as a composition vector (**Int.** ✗). This demonstrates the importance of modeling the interaction between the composition elements not only for the material property prediction task [10, 41], but also for inorganic retrosynthesis planning. **2)** By comparing the methods that utilize retrieval (**Retr.** ✓) with those that do not (**Retr.** ✗), it becomes apparent that using precursor information from reference materials contained in a synthesis literature knowledge base enhances the precursor prediction performance. This underscores the significance of reference materials, which is also a standard practice in the traditional material synthesis process. **3)** We also find that Retrieval-Retro surpasses all baseline models, indicating that our method successfully facilitates inorganic retrosynthesis planning. Furthermore, there is a significant performance enhancement over baseline methods in the year split setting (Table 1 (a)), which is a more realistic and challenging scenario, proving Retrieval-Retro's efficacy in real-world inorganic material synthesis processes.

**Discovering Novel Synthesis Recipes.** As shown in the previous section, the precursor information of reference materials is beneficial for the inorganic retrosynthesis planning of the target material. This observation raises a natural question: *How can we effectively integrate the information from reference materials into synthesis recipes, particularly for novel synthesis recipes?* To answer this question, we evaluate how each component in the model affects the model's capability of deducing novel synthetic recipes for the target material. To do so, we first divide the test set of the year split dataset into **Subset case** and **New case**. As shown in Figure 1, the subset case includes target materials that share a common set of precursors with those in the training set, while the new case comprises target materials that have an entirely new set of precursors. Moreover, we mainly compare Retrieval-Retro with "Graph Network + MPC" in Table 1, since the only difference between the models lies in whether the model directly utilizes the precursor information from reference materials. Then, we separately assess the model's performance in the subset case and the new case.

Table 2: Overall model performance in subset case and new case. "Refer." indicates whether the model explicitly or implicitly uses the precursor information from reference materials.

| Model | Refer. | Retriever | | Subset Case | | | | New Case | | | |
|---|---|---|---|---|---|---|---|---|---|---|---|
| | | MPC | NRE | Top-1 | Top-3 | Top-5 | Top-10 | Top-1 | Top-3 | Top-5 | Top-10 |
| Graph Network | Explicit | ✗ | ✗ | 63.98 (0.34) | 67.95 (0.53) | 68.83 (0.64) | 70.83 (0.81) | 16.37 (1.91) | 22.00 (3.47) | 23.78 (3.48) | 27.93 (1.66) |
| | | ✓ | ✗ | 65.01 (1.10) | 69.06 (1.17) | 69.98 (1.22) | 72.03 (0.92) | 17.63 (1.66) | 22.45 (2.63) | 24.22 (2.65) | 26.22 (2.74) |
| Retrieval-Retro | Implicit | ✓ | ✗ | 65.07 (0.80) | 69.44 (1.27) | 70.41 (1.24) | 72.47 (1.46) | 19.70 (1.08) | 24.52 (1.42) | 26.30 (1.28) | 30.15 (2.05) |
| | | ✓ | ✓ | **66.00** (0.32) | **70.51** (0.61) | **71.76** (0.61) | **73.92** (0.90) | **20.15** (1.29) | **27.04** (1.93) | **28.37** (2.05) | **31.56** (3.44) |

In Table 2, we have following observations: **1)** By comparing the Graph Network with and without the MPC retriever, we find that incorporating the precursor information from reference materials enhances model performance in subset cases. One interesting observation is that, this information negatively impacts performance in Top-10 new case, demonstrating that it can hinder the model's ability to deduce novel synthetic pathways. **2)** Conversely, Retrieval-Retro, which implicitly integrates the precursor information, consistently shows performance improvements in both the subset and new cases, particularly widening the performance gap in the new case—a more realistic and challenging scenario. These findings illustrate the importance of how precursor information from reference materials should be integrated, particularly in identifying new synthetic pathways, which can speed up the material discovery process and reduce synthesis costs. **3)** Interestingly, we find that the NRE retriever also consistently enhances the model performance, a benefit likely stemming from its complementary relationship with the MPC retriever. Specifically, while the MPC retriever captures dependencies between precursors and the target material based on previously observed data, novel synthesis recipes might diverge from the existing synthesis patterns documented in the literature. On the other hand, the NRE retriever employs domain expertise that is independent of these existing patterns, thus filling gaps that the MPC retriever might overlook. By accessing such complementary reference materials, the model is able to acquire additional new precursor information, which contributes to the observed performance improvements. In Table 4, we conduct a qualitative analysis of the materials retrieved by each retriever and examine their impact on model predictions.

In conclusion, we find that each element of Retrieval-Retro plays an effective role in inorganic retrosynthesis planning, particularly in identifying new synthesis recipes, showcasing its potential influence in the field of materials discovery.

## 4.3 Model Analysis

**Ablation Studies: Effects of Retriever.** To verify the effect of the retrievers, we conduct ablation studies by removing the retriever modules. In Table 3, we have following observations: **1)** When reference materials are randomly retrieved without using trained retrievers, the model extracts precursor information that is irrelevant to the synthesis of the target material, leading to a deterioration in performance. **2)** Furthermore, using just one of the retrievers (i.e., either MPC or NRE) leads to underperforms the case when both retrievers are used, showing that the two retrievers are complementary to each other as discussed in Section 4.2. In conclusion, we contend that both MPC and NRE retrievers should be employed to provide informative reference materials to the model, facilitating the extraction of valuable information in inorganic retrosynthesis planning. We provide further ablation studies in Appendix E.3.

**Sensitivity Analysis: Size of Knowledge Base.** We evaluate how the size of the knowledge base affects the model performance. More precisely, from the original database, we sample various sizes of knowledge bases, such as 20%, 40%, and up to 100% (Full) of the size of the original database, and then retrieve reference materials from these sampled subsets. In Figure 4 (a), as expected, the larger the knowledge base, the more accurate the model's predictions. These findings illuminate potential avenues for further development of our model, particularly as more inorganic synthetic recipes are uncovered in the future.

**Sensitivity Analysis: Number of Reference Materials.** Moreover, we investigate how the varying number of reference materials $K$ affects the model performance. In Figure 4 (b), we notice that model performance improves with an increase in the number of references up to a certain point, specifically $K = 3$. This reaffirms the importance of incorporating precursor information from reference materials for inorganic retrosynthesis planning. However, increasing the number of reference materials beyond $K = 3$ does not further enhance model performance, likely due to the introduction of irrelevant or

Table 3: Effect of retrievers on model performance.

| Retriever | Top-K Accuracy | | | | Recall | |
|---|---|---|---|---|---|---|
| | Top-1 | Top-3 | Top-5 | Top-10 | Macro | Micro |
| Random | 58.42 | 63.57 | 64.53 | 66.61 | 58.98 | 59.02 |
| | (1.68) | (0.64) | (0.53) | (0.64) | (0.64) | (0.64) |
| MPC only | 59.96 | 64.49 | 65.57 | 68.14 | 59.57 | 59.64 |
| | (0.66) | (0.74) | (0.96) | (0.81) | (0.67) | (0.69) |
| NRE only | 60.28 | 64.70 | 65.75 | 68.00 | 59.88 | 59.95 |
| | (0.63) | (1.21) | (1.17) | (1.39) | (0.63) | (0.60) |
| Retrieval-Retro | 61.16 | 65.92 | 67.18 | 69.45 | 60.97 | 61.06 |
| (MPC + NRE) | (0.38) | (0.71) | (0.76) | (1.03) | (0.62) | (0.62) |

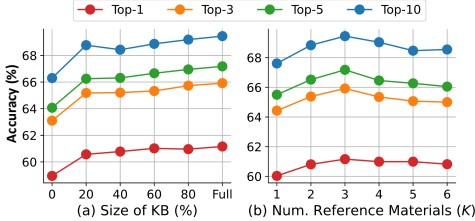

Figure 4: Sensitivity Analysis results. KB refers to the knowledge base.

noisy precursor information that does not pertain to the target material. Thus, extracting precursor information from a suitable number of retrieved materials is essential for optimal performance.

**Qualitative Analysis.** We present a qualitative analysis of our proposed method in retrosynthesis planning for the target material $Pb_9[Li_2(P_2O_7)_2(P_4O_{13})_2]$, which can be synthesized from the precursor set: $\{Li_2CO_3, NH_4H_2PO_4, PbO\}$. As shown in Table 4, when only the MPC retriever is used, the method fails to predict the entire precursor set due to insufficient extraction of precursor information from the retrieved materials. However, when the NRE retriever is used alongside the MPC retriever, our method successfully predicts the complete precursor set for the target material. This success is attributed to the NRE retriever, which complements the model by providing complementary retrieved materials whose precursors are feasible for synthesizing the target material, thereby enabling the extraction of diverse precursor information. For instance, the NRE retriever allows our method to extract precursor information from $Pb_3(PO_4)_2$, which contains the essential precursor PbO, a direct precursor for the target material. Due to the complementary nature of the retrievers, our method can effectively extract precursor information from informative reference materials, leading to enhanced predictions.

Table 4: Qualitative Analysis ( Target Material: $Pb_9[Li_2(P_2O_7)_2(P_4O_{13})_2]$ ).

| Model | Retriever | Retrieved Material | Corresponding Precursor Sets | Predicted Precursor Set | Answer |
|---|---|---|---|---|---|
| Only MPC | MPC | LiNaPbPO $Li_{0.5}Na_{0.5}PO_3$ $Li_3V_{1.92}Al_{0.08}(PO_4)_3$ | $\{Li_2CO_3, H_3PO_4, Na_2CO_3, Pb_3O_4\}$ $\{Li_2CO_3, NH_4H_2PO_4, NaPO_3\}$ $\{Al, V_2O_5, LiH_2PO_4\}$ | $\{Li_2CO_3, NH_4H_2PO_4\}$ | ✗ |
| MPC + NRE (Ours) | MPC | LiNaPbPO $Li_{0.5}Na_{0.5}PO_3$ $Li_3V_{1.92}Al_{0.08}(PO_4)_3$ | $\{Li_2CO_3, H_3PO_4, Na_2CO_3, Pb_3O_4\}$ $\{Li_2CO_3, NH_4H_2PO_4, NaPO_3\}$ $\{Al, V_2O_5, LiH_2PO_4\}$ | $\{Li_2CO_3, NH_4H_2PO_4, PbO\}$ | ✓ |
| | NRE | $Pb_3(PO_4)_2$ $Li_3P$ $PbP_7$ | $\{PbO, NH_4H_2PO_4\}$ $\{P, Li\}$ $\{P, Pb\}$ | | |

## 5 Related Works

### 5.1 Machine Learning for Inorganic Retrosynthesis

Owing to the intricate nature of inorganic retrosynthesis, the production of inorganic materials entails a mix of numerous synthetic variables, leading to a variety of subtasks: 1) Learning favorable reaction pathways with thermodynamic variables when a target and precursor set are provided [2, 27, 35], 2) Employing regression models to predict conditions of the reaction pathway, such as heating temperature and time [13], and 3) Predicting possible precursor sets for a target material. In this work, we focus on the precursor selection task, which aims to predict feasible sets of precursors for synthesizing a target material [17, 18, 12].

**Precursor Prediction Task.** As a pioneering work, CVAE [17] generates synthesis actions and precursors through a generative model, i.e., conditional variational autoencoder [19]. However, it frequently produces chemically invalid precursors by relying on the generative model. ElemwiseRetro [18] alleviates the issue by formulating the precursor prediction task as a multi-class classification with dozens of manually created precursor templates. While it successfully suggests chemically valid precursors, it fails to incorporate knowledge from synthesis literature, which is common practice in the field of material science. More recently, inspired by the chemists' synthesis practice, He et al. [12] propose to use the precursor information of materials that are similar to the target material, which is retrieved from the related literature. This approach utilizes the precursors of the retrieved material as explicit conditions for predicting the precursors of the target material. However, such

explicit utilization of precursor information can inhibit the model's capability in providing novel synthetic routes for target material. Different from the previous work, we propose to implicitly utilize precursor information of similar material rather than directly using it.

## 5.2 Composition-based representation learning for Inorganic materials

While most machine learning approaches for inorganic materials predominantly utilize structural information, in real-world material discovery, structural data is often unavailable due to the high costs of computational resources. Consequently, some ML methods opt to use compositional information of inorganic materials instead of structural details. For example, ElemNet [15] proposes to learn the representation of material with compositional information using deep neural networks (DNNs). Roost [10] learns material representation by building fully connected graphs based on the composition to model interactions between elements by graph neural networks (GNNs). Additionally, CrabNet [41] successfully applies a self-attention mechanism to the element-derived matrix to accurately predict the properties of materials. Although these methods were originally designed for predicting material properties, they can also be applied to inorganic retrosynthesis as they similarly rely on the composition information of materials.

## 5.3 Difference between Inorganic Retrosynthesis and Organic Retrosynthesis

Both organic and inorganic retrosynthesis are challenging tasks that predict the synthesis of materials by breaking down the target material into simpler precursors. However, there are significant differences between organic and inorganic retrosynthesis. Organic retrosynthesis [45, 40, 7, 9, 33] deals with organic compounds, which are molecules primarily composed of elements such as carbon, hydrogen, oxygen, nitrogen, and sulfur. These compounds are represented using molecular structure graphs or SMILES strings. In contrast, inorganic retrosynthesis involves inorganic compounds, which can include a wider variety of elements, often including metals, and have structures that periodically repeat in unit cells. Another key difference lies in the use of structural information during retrosynthesis planning. Organic retrosynthesis utilizes structural information such as functional groups and reaction centers of organic compounds, which indicate the properties of a material and its reactivity with other molecules, to predict simpler molecules (precursors) into which the target molecule can be broken down. Inorganic compounds, however, have relatively unexplored generalized synthesis mechanisms compared to organic compounds, and calculating their structures is expensive. Therefore, it is challenging to directly use structural information for retrosynthesis planning. Instead, inorganic retrosynthesis [12, 17, 18] often relies solely on the chemical composition of the materials, distinguishing it from organic retrosynthesis.

# 6 Limitations

We aim to identify favorable precursor sets by considering the thermodynamic relationships between materials and their precursor set. However, in the actual synthesis process, the phase changes of materials are influenced by synthesis temperature, synthesis time, pressure condition and pairwise reactions between precursors. Taking these factors into account would enable more accurate precursor set predictions. Nevertheless, in situations where experimental data (such as temperature and pressure) are unavailable, we estimate the reaction energy solely from the formation energy calculated under consistent temperature and pressure conditions using a trained predictor, derived from the composition of materials. Considering multiple synthesis conditions can lead to more precise predictions of precursor sets.

# 7 Conclusion

In this study, we introduce Retrieval-Retro, a novel method for inorganic retrosynthesis planning by extracting the precursor information of retrieved reference material implicitly. To do so, we employ various attention layers that enhance and extract the information from the reference material. Moreover, we design a neural reaction energy (NRE) retriever that provides complementary reference material to the MPC retriever, allowing Retrieval-Retro to integrate precursor information from a broader range of reference materials through domain expertise. Through extensive experiments, including assessments in realistic scenarios, we demonstrate the effectiveness of implicit extraction of precursor information and NRE retriever in discovering novel synthesis recipes of target material, demonstrating the potential impact of Retrieval-Retro in the field.

## Acknowledgements

This study was supported by Korea Research Institute of Chemical Technology (No.: KK2351-10), the National Research Foundation of Korea(NRF) grant funded by the Korea government(MSIT) (RS-2024-00406985), and NRF grant funded by Ministry of Science and ICT (NRF-2022M3J6A1063021).

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

# Supplementary Material for
# Retrieval-Retro: Retrieval-based Inorganic Retrosynthesis with Expert Knowledge

# A Implementation Details

In this section, we provide implementation details of Retrieval-Retro.

## A.1 Composition Graph Encoder

As a composition graph encoder, we mainly consider graph network [3], which is the generalized version of various graph neural networks. Building on prior research, our graph neural networks are divided into two components: the encoder and the processor. The encoder is responsible for learning the initial representation of elements, while the processor manages message passing between elements. To describe this more formally, for an element $e_i$ and the edge $a_{i,j}$ connecting element $e_i$ to element $e_j$, the node encoder $\phi_{node}$ and the edge encoder $\phi_{edge}$ generate initial representations of the element as follows:

$$\mathbf{e}_i^0 = \phi_{node}(\mathbf{e}_i), \quad \mathbf{a}_{ij}^0 = \phi_{edge}(\mathbf{a}_{ij}), \tag{5}$$

where $\mathbf{e}_i$ is the initial feature of element $i$ and $\mathbf{a}_{ij}$ is the concatenated feature of element $i$ and $j$, i.e., $\mathbf{a}_{ij} = (\mathbf{e}_i \| \mathbf{e}_j)$. Using the initial representations of elements and edges, the processor is designed to facilitate message passing among the elements and to update the representations of both elements and edges in the following manner:

$$\mathbf{a}_{ij}^{l+1} = \psi_{edge}^l(\mathbf{e}_i^l, \mathbf{e}_j^l, \mathbf{a}_{ij}^l), \quad \mathbf{e}_i^{l+1} = \psi_{node}^l(\mathbf{e}_i^l, \sum_{j \in \mathcal{E}} \mathbf{a}_{ij}^{l+1}), \tag{6}$$

where $\mathcal{E}$ represents element set comprising the material, and $\psi$ is a two-layer MLP with non-linearity. Note that we use three message passing layers within the processor, i.e., $l = 0, \ldots, 2$.

## A.2 Training Details

**Model Training.** Our method is implemented on Python 3.8.13, and Torch-geometric 2.0.4. In all our experiments, we use the AdamW optimizer for model optimization. We train the model for 500 epochs across all tasks, while the model is early stopped if there is no improvement in the best validation Top-5 Accuracy for 30 consecutive epochs. All experiments are conducted on a 48 GB NVIDIA RTX A6000.

**Hyperparameters.** We have detailed the hyperparameter specifications in the table 5. For Retrieval-Retro, we adjust the hyperparameters within specific ranges as follows: number of message passing layers in GNN $L'$ in {2, 3}, number of cross-attention layers $C$ in {1,2}, size of hidden dimension $D$ in {256}, number of self-attention layers $S$ in {1,2}, learning rate $\eta$ in {0.0001, 0.0005, 0.001}, batch size $B$ in {32, 64, 128} and number of retrieved materials $K$ in {1,2,3,4,5,6}. We present the test performance based on the best results obtained on the validation set.

Table 5: Hyperparameter specifications of Retrieval-Retro.

| Hyperparameters | Dataset Split | |
|---|---|---|
| | Year Split | Random Split |
| # Message Passing Layers ($L'$) | 3 | 3 |
| # Self-Attention Layers ($S$) | 1 | 1 |
| # Cross-Attention Layers ($C$) | 2 | 2 |
| Hidden Dim. ($D$) | 256 | 256 |
| Learning Rate ($\eta$) | 0.0001 | 0.0001 |
| Batch Size ($B$) | 128 | 32 |
| # of Reference Materials ($K$) | 3 | 3 |

# B Dataset

In this section, we provide further details on the dataset used for experiments.

- Following previous study[12], we use 33,343 inorganic solid-state synthesis recipes extracted from 24,304 materials science papers [20] obtained from the github repository [2] of previous work [12]. Following the preprocessing step of previous work, we obtain a total number of 28,598 target materials, which have a diverse number of possible precursor sets as shown in Table 6. In other words, a single target material can have multiple corresponding ground-truth precursor sets. Furthermore, in our experiments, we exclude materials from the validation and test sets that contain precursors not present in the training set, as the classifier would not be trained on those absent precursors. Consequently, for the year split, we use 24,034 entries for the training set, 1,842 for the validation set, and 2,558 for the test set. For the random split, the number of target materials varies across five different trials due to the differing compositions of the training, validation, and test sets in each trial.

Table 6: The distribution of data based on the number of ground truth precursor sets.

| # Precursor sets | 1 | 2 | 3 | 4 | 5 | 6 | 7 | 8 | 9 | 10 | 11 | 12 | 13 | 14 | 17 | 23 | 28 | 29 | 34 | Total |
|---|---|---|---|---|---|---|---|---|---|---|---|---|---|---|---|---|---|---|---|---|
| # Data | 26,944 | 1,168 | 257 | 108 | 42 | 32 | 13 | 8 | 7 | 5 | 4 | 1 | 2 | 2 | 1 | 1 | 1 | 1 | 1 | 28,598 |

In Section 3.1, we propose to pre-train the Neural Reaction Energy (NRE) retriever with density functional theory (DFT) calculated formation energy and then fine-tune to experimental data.

- For DFT-calculated data, we use **Materials Project** [14] database [3], which is an openly accessible database that provides various material properties calculated using DFT. From the database, we have collected 80,162 unique compositions along with their respective formation energies. It is important to note that when a composition is associated with multiple structures and formation energies, we only consider the lowest formation energy for the material, as it is the most likely to exist.

- For experimental data [32], we download experimental formation energy data from previous work's repository[16][4], which aim to develop robust material property prediction models via transfer learning. We then filter the data down to 1,637 entries using the same preprocessing methods as those applied to DFT-calculated data.

- For calculating the Gibbs free energy, we use the formation energy from the Materials Project that we used is calculated at 0 K and 0 atm using DFT and the experimental formation energy is taken from [16], which reports measurements at 298.15 K and 1 atm.

## C   Baseline Methods

In this section, we provide detailed explanations of the baseline methods compared in Section 4. We first provide details on the two previous inorganic retrosynthesis planning methods as follows:

- **He et al. [12]** introduces a new approach to inorganic retrosynthesis planning by retrieving reference materials that share similar properties with the target material. Initially, it proposes using a masked precursor completion (MPC) retriever, described in detail in Section 3.1, which depends solely on the composition vector $\mathbf{x}$ of the inorganic material.

- **ElemwiseRetro** [18] introduces a method for inorganic retrosynthesis planning that represents the target material as a fully connected composition graph and predicts the likelihood of various precursor sets from the available precursor templates.

Furthermore, as outlined in Section 5.2, there are existing studies that develop material representations based on the composition information of inorganic materials. Although these studies were originally aimed at predicting material properties, we have adapted the prediction heads to make them suitable for inorganic retrosynthesis planning. Here, we provide details on the methods as follows:

- **Roost** [10] suggests employing GNNs to learn representations of inorganic materials by representing their composition as a fully connected graph, with nodes representing the unique elements within the

---

[2]https://github.com/CederGroupHub/SynthesisSimilarity
[3]https://materialsproject.org/
[4]https://github.com/wolverton-research-group/qmpy/blob/master/qmpy/data/thermodata/ssub.dat

composition. This enables the model to explore the complex relationships between the constituent elements, thus capturing physically significant properties and interactions.

- **CrabNet** [41] proposes to model the complex interaction between constituent elements with a Transformer self-attention mechanism [39] to adaptively learn the representation of elements based on their chemical environment.

In addition to existing studies, we introduce further baseline models that can support our claims as described below:

- **Composition MLP** aims to predict the set of precursors based on the composition vector $\mathbf{x}$ of the inorganic material as follows:
$$\hat{\mathbf{y}} = \text{MLP}(\mathbf{x}). \tag{7}$$
  The sole distinction between **Composition MLP** and He et al. [12] is whether the model retrieves reference materials, highlighting the effectiveness of the retrieval mechanism in inorganic retrosynthesis planning.

- **Graph Network** [3] aims to predict the set of precursors based on the composition graph $\mathcal{G}$ of the inorganic material as follows:
$$\hat{\mathbf{y}} = \text{Graph Network}(\mathcal{G}). \tag{8}$$

- **Graph Network + MPC** improves upon He et al. [12] by replacing its material encoder, originally an MLP that processed the composition vector $\mathbf{x}$, with a graph network. Since the primary distinction between **Graph Network** and **Graph Network + MPC** is whether the model retrieves reference materials, comparing these methods allows us to evaluate the effectiveness of using reference materials in inorganic retrosynthesis planning. Furthermore, given that **Graph Network** serves as the backbone architecture for Retrieval-Retro, comparing **Graph Network + MPC** with Retrieval-Retro enables the assessment of the effectiveness of the implicit fusion and NRE retriever.

## D  Evaluation Protocol

We use Top-K exact match accuracy, Macro recall, and Micro Recall for evaluating the capability of Retrieval-Retro in the inorganic retrosynthesis task. To begin evaluation, we first define the number of precursors of each material. Following many multi-label classification works [5, 47], the precursor labels are considered to be positive if their probabilities $\hat{y}_i$ are greater than 0.5. We set the number of positive precursor labels, $L$ as the number of precursor that each material has.

**Top-K exact match accuracy.** For Top-K exact match accuracy, we select K sets which has $L$ precursors, where each set is chosen based on the highest product of probabilities of its $L$ precursors. Then, for each of the K sets, if there exists a set that exactly matches the correct precursor set, it is scored as 1; otherwise, it is scored as 0. The average is then calculated over all test materials.

**Micro and Macro Recall.** Note that a material can have a varied number of possible precursor sets, as shown in Table 6. We use Micro and Macro Recall to evaluate such cases. We select the same number of precursor sets as the material has in the same way as we evaluate for Top-K exact match accuracy. We then calculated the macro recall by comparing each obtained set with the correct sets for each material. If a set matched exactly, it was scored as 1; otherwise, it was scored as 0. These scores were averaged for each test material. For micro recall, we calculated the overall average across all test materials.

## E  Additional Experiments

### E.1  Performance on Various Backbone Architecture

Since Retrieval-Retro is agnostic to the various composition graph encoder architectures, we validate the effectiveness of Retrieval-Retro within a variety of GNN architectures. In Figure 5, we observe performance improvements across all GNN architectures tested, confirming that Retrieval-Retro framework can consistently improve vanilla GNN architecture for inorganic retrosynthesis planning.

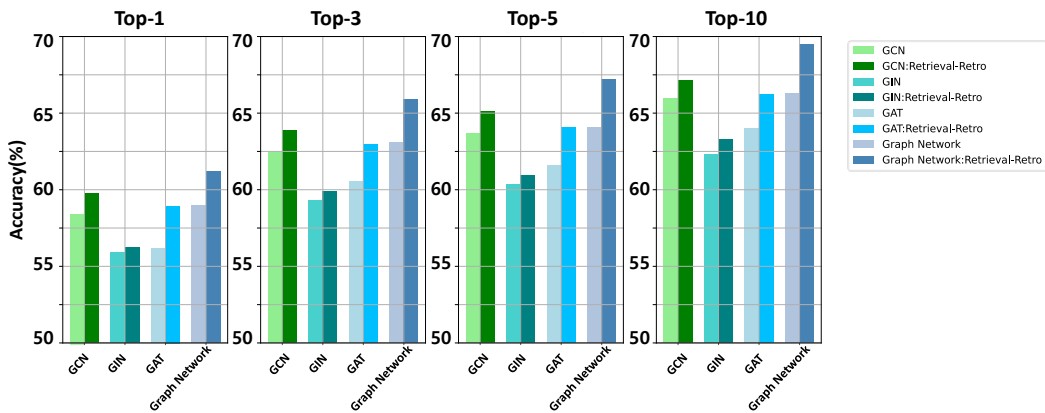

Figure 5: Performance Improvements across Various GNN Backbones

## E.2 Neural Reaction Retriever

As described in Section 3.1, we develop a composition-based formation energy predictor tailored for experimental data. To do so, we initially pre-train the GNN predictor on extensive DFT-calculated data [14] and then fine-tune on experimental formation energy data [32]. In this section, we demonstrate whether the pre-train and fine-tune strategy is effective in predicting formation energies. To do so, we divide the entire DFT dataset into three parts: 80% for training, 10% for validation, and 10% for testing. The model is trained over 1,000 epochs, employing early stopping if there is no improvement in the validation loss for 50 consecutive epochs. Similarly, we partition the experimental dataset into 80/10/10% splits, following the same training approach as for the DFT data. While our model begins training experimental data using pre-trained checkpoints from a model trained on DFT data, we also conduct comparisons with a model trained exclusively on experimental data (Ablation in Table 7). Table 7 shows that pre-training with a DFT-calculated dataset enhances model performance, indicating that the NRE retriever efficiently retrieves reference materials related to formation energies.

Table 7: Formation energy predictor performance on experimental formation energy data.

|  | DFT | Exp. | MAE (eV/atom) |
|---|---|---|---|
| Ablation | ✗ | ✓ | 0.0761 |
| Ours | ✓ | ✓ | **0.0643** |

## E.3 Ablation Study on Attention Layer

We conduct an ablation study on the self-attention layer in Retrieval-Retro. By removing the self-attention layer, Retrieval-Retro extracts implicit precursor information by integrating the representation of the target material without the enhanced reference materials through a cross-attention mechanism. As shown in Table 8, this results in a noticeable decline in model performance, underscoring the importance of extracting precursor information from the material's enhanced representation via self-attention. However, even with just the cross-attention mechanism, Retrieval-Retro still outperforms the basic Graph Network.

Table 8: Effect of attention layers on model performance.

| Model | Attention Layers | | Year Split | | | |
|---|---|---|---|---|---|---|
| | Self-Attention | Cross-Attention | Top-1 | Top-3 | Top-5 | Top-10 |
| Graph Network | ✗ | ✗ | 58.95 (0.41) | 63.10 (0.63) | 64.07 (0.68) | 66.30 (0.62) |
| Retrieval-Retro | ✗ | ✓ | 60.34 (0.34) | 65.08 (0.58) | 66.40 (0.60) | 68.99 (0.60) |
| Retrieval-Retro | ✓ | ✓ | **61.16** (0.38) | **65.92** (0.71) | **67.18** (0.76) | **69.45** (1.03) |

### E.4 Model Training and Inference Time

In this section, we provide time and inference analysis to verify the efficiency of Retrieval-Retro , as shown in Table 9. We observe that methods using retrievers have slightly longer training times compared to those that do not use retrievers. However, since we preprocess the materials to retrieve using a pretrained retriever prior to training, the training time for the model utilizing retrievers remains manageable. Additionally, Roost and CrabNet exhibit significantly longer training times compared to other methods, which can be attributed to their distinctive modeling interaction mechanisms.

Table 9: Training and inference time per epoch (sec/epoch).

| Model | Retriever | Training Time | Inference Time |
|---|---|---|---|
| Composition MLP | ✗ | 0.4129 | 0.0007 |
| He et al. [12] | ✓ | 1.6622 | 0.0103 |
| ElemwiseRetro | ✗ | 0.7528 | 0.0012 |
| Roost | ✗ | 2.6382 | 13.6702 |
| CrabNet | ✗ | 9.7238 | 0.1123 |
| Graph Network | ✗ | 0.6917 | 0.0019 |
| Graph Network + MPC | ✓ | 2.4361 | 0.0107 |
| Retrieval-Retro | ✓ | 3.2601 | 0.0116 |

## F Broader Impacts

**Potential Positive Scientific Impacts.** Our work, Retrieval-Retro, explores the automation of precursor prediction, a process traditionally dependent on a chemist's expertise. We have developed a model that learns novel synthesis recipes by implicitly extracting precursor information from retrieved materials. This enables Retrieval-Retro to effectively recommend precursor sets for target materials and facilitates its use in real autonomous material synthesis processes [35].

**Potential Negative Societal Impacts.** Although this work demonstrate good predictive capabilities for precursor prediction, it lacks uncertainty estimation. Therefore, it is essential to use this model collaboratively with chemists for effective precursor prediction.

## G Pseudo Code

---

**Algorithm 1:** Pseudocode of Retrieval-Retro.

---

**Input :** Composition based fully connected graph $\mathcal{G} = (\mathbf{X}, \mathbf{A})$, Retrieved reference material graph from MPC retriever $\mathcal{G}_{\text{r: MPC}} = (\mathbf{X}_{\text{r: MPC}}, \mathbf{A}_{\text{r: MPC}})$, Retrieved reference material sets from NRE retriever $\mathcal{G}_{\text{r: NRE}} = (\mathbf{X}_{\text{r: NRE}}, \mathbf{A}_{\text{r: NRE}})$, Number of self attention layer $s$, Number of cross attention layer $c$, The number of Retrieved Material $K$, Graph Network GNN

1 $\mathbf{g}_t \leftarrow \text{GNN}(\mathbf{X}, \mathbf{A})$
2 **for** $i = 1$ *to* $K$ **do**
3 $\quad \mathbf{g}^i_{r:MPC} \leftarrow \text{GNN}(\mathbf{X}_{\text{r: MPC}}, \mathbf{A}_{\text{r: MPC}})$
4 **end**
5 $\mathbf{G}'^s_r \leftarrow \text{Self-Attention}(\mathbf{g}'^k_r, s)$
6 $\mathbf{g}^c_t \leftarrow \text{Cross-Attention}(\mathbf{g}'^{c-1}_r, \mathbf{G}'^s_r, c)$ `// Repeat above process line 2 - 6 for NRE retriever`
7 $\hat{\mathbf{y}} \leftarrow \phi_{\text{classifier}}(\mathbf{g}_t || \mathbf{g}^C_{t:\text{MPC}} || \mathbf{g}^C_{t:\text{NRE}})$
8 $\mathcal{L} \leftarrow \text{Binary Cross Entropy}(\hat{\mathbf{y}}, \mathbf{y})$

---

---

**Algorithm 2:** Pseudocode of MPC Retriever.

---

**Input:** A chemical composition of material $\mathbf{x}$, A learnable precursor embeddings $\mathbf{P}$, An embedding for the $i$-th precursor $\mathbf{p}_i$ of $\mathbf{P}$, Ground truth Precursor $\mathbf{y}$, MLP $\mathbf{M}$, Knowledge BaseKB

1  $\mathbf{m} \leftarrow \mathbf{M}(\mathbf{x})$

2  Construct a perturbed $\tilde{\mathbf{P}}$ by applying randomly perturbed $\tilde{\mathbf{y}}$ as a mask. ($\tilde{\mathbf{p}}_i$ is masked if $\tilde{y}_i = 0$, and is left unchanged otherwise.) `// Masking embeddings P`

3  $\mathbf{s} \leftarrow \text{Cross-Attention}(\mathbf{m}, \tilde{\mathbf{P}}, \tilde{\mathbf{P}})$

4  $\hat{\mathbf{y}} \leftarrow \sigma(\mathbf{s}^\top \mathbf{p}_i)$ `// Calculate the probability of each precursor`

5  $\mathcal{L}_{Training} \leftarrow \text{Binary Cross Entropy}(\hat{\mathbf{y}}, \mathbf{y})$

6  Calculate the cosine similarity between the target material and all materials in the KB using $\mathbf{M}$

7  Retrieve the top $K$ materials and save the retrieved material sets. `// Construct retrieved material Sets`

---

---

**Algorithm 3:** Pseudocode of NRE Retriever.

---

**Input:** Composition based fully connected graph (material from DFT calculated data) $\mathcal{G}_{DFT} = (\mathbf{X}_{DFT}, \mathbf{A}_{DFT})$, Composition based fully connected graph (material from experimental data) $\mathcal{G}_{Exp} = (\mathbf{X}_{Exp}, \mathbf{A}_{Exp})$, Ground truth Formation energy from DFT calculated data $\mathbf{H}_{DFT}$, Ground truth Formation energy from experimental data $\mathbf{H}_{Exp}$, Graph Network GNN, Knowledge BaseKB.

1  $\hat{\mathbf{H}}_{DFT} \leftarrow \text{GNN}(\mathbf{X}_{DFT}, \mathbf{A}_{DFT})$

2  $\mathcal{L}_{Pre-train} \leftarrow \text{MAE}(\hat{\mathbf{H}}_{DFT}, \mathbf{H}_{DFT})$ `// Pre-train using DFT calculated data`

3  Initialize the GNN with weigths of pre-trained GNN

4  $\hat{\mathbf{H}}_{Exp} \leftarrow \text{GNN}(\mathbf{X}_{Exp}, \mathbf{A}_{Exp})$

5  $\mathcal{L}_{Fine-tune} \leftarrow \text{MAE}(\hat{\mathbf{H}}_{Exp}, \mathbf{Form.E}_{Exp})$ `// Fine-tune using Experimental data`

6  Calculate the formation energy ($\mathbf{H}$) of the target material in the dataset and the precursor set of all materials in the knowledge base (KB) using a fine-tuned GNN. `// Calculate` $\Delta H$

7  Calculate $\Delta G$ following $\Delta G \approx \Delta H = H_{Target} - H_{Precursor\ set}$ `// Calculate` $\Delta G$

8  Retrieve $K$ materials with most negative $\Delta G$ from KB per target material, then save the retrieved material sets. `// Construct Retrieved Material Sets`

---

