# OpenReview forum: "Retrieval-Retro: Retrieval-based Inorganic Retrosynthesis with Expert Knowledge"
_NeurIPS.cc/2024/Conference — NeurIPS 2024 poster_

### Official Review · Reviewer_X1nP · 2024-07-08

**Soundness:** 3
**Presentation:** 3
**Contribution:** 3
**Rating:** 6
**Confidence:** 2

**Summary:**

I'm not an expert in this field. I'm familiar with organic retrosynthesis prediction but not familiar with inorganic retrosynthesis planning.

This paper first trains a retriever to determine which materials to reference. Then this paper trains a model for material selections.

**Strengths:**

1. This inorganic retrosynthesis planning is different from organic retrosynthesis planning. The proposed method is novel in my opinion. Inorganic retrosynthesis planning operates step by step while this paper proposes the material within one step.

2. Writing is clear. I can follow this paper.

3. The proposed method achieves good performance.

**Weaknesses:**

1. Little discussion on the difference between inorganic retrosynthesis planning and organic retrosynthesis planning.

2. Little discussion on computational complexity and the space of the material.

**Questions:**

N/A

---

> ### Author Rebuttal · Authors · 2024-08-07
>
> Thank you for your valuable comments and for acknowledging the novelty of our work in inorganic retrosynthesis! We are more than willing to address any questions in detail.
>
> ---
> **[W1]**
>
> As the reviewer suggested, describing the differences between organic retrosynthesis and inorganic retrosynthesis would make the manuscript easier to understand. Thank you for the valuable feedback. We plan to add this section to the manuscript in the future.
> Both organic and inorganic retrosynthesis are challenging tasks that predict the synthesis of materials by breaking down the target material into simpler precursors. However, there are significant differences between organic and inorganic retrosynthesis.
>
> Organic retrosynthesis deals with organic compounds, which are molecules primarily composed of elements such as carbon, hydrogen, oxygen, nitrogen, and sulfur. These compounds are represented using molecular structure graphs or SMILES strings. In contrast, inorganic retrosynthesis involves inorganic compounds, which can include a wider variety of elements, often including metals, and have structures that periodically repeat in unit cells.
>
> Another key difference lies in the use of structural information during retrosynthesis planning. Organic compounds utilize structural information such as functional groups and reaction centers, which indicate the properties of the material and its reactivity with other molecules, to predict simpler molecules (precursors) into which the target molecule can be broken down. Inorganic compounds, however, have relatively unexplored generalized synthesis mechanisms compared to organic molecules, and calculating their structures is expensive. Therefore, it is challenging to directly use structural information for retrosynthesis planning. Instead, inorganic retrosynthesis often relies solely on the chemical composition of the materials, distinguishing it from organic retrosynthesis.
>
> ---
>
> **[W2]**
>
> RetroPLEX initially trains two retrievers in advance, the MPC retriever and the NRE retriever, to simplify the complexity of the model by leveraging the materials retrieved from these trained retrievers. The maximum GPU memory required to train the RetroPLEX model is less than 4.5GB, so it is also possible to train the model on GPUs with smaller memory capacities, such as the NVIDIA GeForce Titan (12GB). We also measured the training time by  training the model using only the CPU, and found that it takes 22.02 seconds per epoch, which is approximately 7 times longer than the 3.26 seconds per epoch when using the GPU. Since the model typically converges in around 150 epochs, the training time is manageable. Consequently, even with the notably longer training duration, it is still feasible to train the model in environments without a GPU.
>
> We use a dataset of 33,343 inorganic solid-state synthesis records extracted from 24,304 material science papers [1]. After preprocessing, we utilize data for 28,434 target materials. Since the synthesis data was extracted from material science publications up to 2020, allowing us to consider relatively recent target materials, and given that this is the largest dataset among available material synthesis datasets, we believe it adequately captures the practices of actual material synthesis literature.
>
> [1] Kononova, Olga, et al. "Text-mined dataset of inorganic materials synthesis recipes." Scientific data 6.1 (2019): 203.

---

> ### Author Response · Authors · 2024-08-11
> **Gentle reminder for author reviewer discussion**
>
> Thank you once again for your valuable review and feedback. We kindly ask if you could please take a moment to confirm whether our rebuttal has adequately addressed your comments and concerns. Thank you for your consideration.

---

> > ### Comment · Reviewer_X1nP · 2024-08-14
> >
> > I acknowledge the response from authors and decide to maintain my score.

---

### Official Review · Reviewer_URJP · 2024-07-11

**Soundness:** 2
**Presentation:** 3
**Contribution:** 2
**Rating:** 5
**Confidence:** 1

**Summary:**

This paper lies in the domain of AI for chemistry, and this paper proposes RetroPLEX for inorganic retrosynthesis planning. The proposed approach is comprised of two components: masked precursor completion retriever and neural reaction energy retriever.

**Strengths:**

The writing is mostly clear.

The experiment results are thoroughly discussed.

**Weaknesses:**

The model training subsection has not detailedly elaborated on the training process of the proposed approach. The authors are encouraged to provide the pseudocode of the proposed approach.

**Questions:**

Can the proposed approach provide some insights into the AI methods for other domains, e.g., robotics and mixed integer linear programming?

**Limitations:**

The authors have discussed the potential limitations in the Appendix.

---

> ### Author Rebuttal · Authors · 2024-08-07
>
> Thank you for your valuable comments on our work! We are more than willing to address both the weakness and the question in detail.
>
> ---
>
> **[W1]**
>
> While we have endeavored to thoroughly explain the training process of RetroPLEX, the limited space of the submission may not have allowed for complete clarity. At the reviewer's request, we have included the model's pseudocode in the attached PDF and will ensure it is incorporated into the Appendix in our final submission. We apologize for any confusion this may have caused and are happy to provide detailed answers if further clarification is needed.
>
> ---
>
> **[Q1]**
>
> The core idea of our proposed approach is to utilize a retriever tailored to the task at hand and to leverage the retrieved information through attention mechanisms. From the perspective of using retrievers, in robotics domains such as motion planning and autonomous navigation, appropriate information matching the situation and task can be effectively utilized through a retriever from a database. When using large language models (LLMs) for robotics control, the intrinsic issues of hallucination and lack of updated information in LLMs can be effectively handled through an online-retrieval approach. Furthermore, when solving a problem, if we can break it down into step-by-step instructions and retrieve appropriate action patterns for each step, we can expect satisfactory task performance compared to not using retrieval.

---

> ### Author Response · Authors · 2024-08-11
> **Gentle reminder for author reviewer discussion**
>
> Thank you once again for your valuable review and feedback. We kindly ask if you could please take a moment to confirm whether our rebuttal has adequately addressed your comments and concerns. Thank you for your consideration.

---

### Official Review · Reviewer_wQcK · 2024-07-12

**Soundness:** 3
**Presentation:** 3
**Contribution:** 3
**Rating:** 5
**Confidence:** 3

**Summary:**

The manuscript presents a approach, RetroPLEX, for inorganic retrosynthesis planning. The authors propose RetroPLEX, a method that implicitly extracts precursor information from reference materials using attention layers. Additionally, they incorporate domain expertise by considering the thermodynamic relationships between target materials and potential precursors. The manuscript presents a approach, RetroPLEX, for inorganic retrosynthesis planning. The authors propose RetroPLEX, a method that implicitly extracts precursor information from reference materials using attention layers. Additionally, they incorporate domain expertise by considering the thermodynamic relationships between target materials and potential precursors.

**Strengths:**

1. The idea of implicitly extracting precursor information from reference materials using retrieval and attention mechanisms is interesting and potentially beneficial for discovering new synthesis recipes.
2. The incorporation of thermodynamic relationships (∆G) through the NRE retriever is a useful addition, reflecting domain knowledge in inorganic synthesis.
3. The paper includes a variety of experiments comparing RetroPLEX to existing methods.

**Weaknesses:**

1. The method is pretty complex, including two retrievers, MPC and NRE. The complex structure, while powerful, could pose challenges in terms of computational efficiency and scalability, especially when deployed in resource-limited settings.

2. The manuscript lacks clarity in several areas. For example, the explanation of the NRE retriever's training process is unclear.

**Questions:**

1.  Why were the MPC and NRE retrievers chosen? How do they complement each other?

2. Given the apparent complexity of the proposed model, what are the computational requirements? Can the authors provide a detailed analysis of the time and resource requirements for training and using RetroPLEX?

3. How does the model's performance scale with increasing dataset sizes or computation complexity?

4. Can the authors provide qualitative examples of the importance of each retriever? For example, provide examples where each retriever plays a crucial role in identifying relevant reference materials or extracting precursor information.

**Limitations:**

The authors have discussed the limitations.

---

> ### Author Rebuttal · Authors · 2024-08-07
>
> Thank you for your valuable comments on our work! We are more than willing to address each of the specific weaknesses and questions in a detailed manner.
>
> ---
>
> **[W1&Q2]**
>
> Indeed, we have provided the complexity of the model in terms of model training and inference in Appendix E.5. Upon the reviewer’s request, we provide further details in terms of GPU memory and computation resources. Specifically, we used a CPU Intel Xeon Gold 6326 and a single NVIDIA GeForce A6000 (48GB) GPU. The maximum GPU memory required to train the RetroPLEX model is less than 4.5GB, so it is also possible to train the model on GPUs with smaller memory capacities, such as the NVIDIA GeForce Titan (12GB). We also measured the training time by  training the model using only a single CPU, and found that it takes 22.02 seconds per epoch, which is approximately 7 times longer than the 3.26 seconds per epoch when using a single GPU. Despite the longer training time when using a CPU, we validated that it is still possible to train the model in environments without a GPU.
> Additionally, despite the apparent complexity of RetroPLEX, we predefine the reference materials for each target material before training. Consequently, as demonstrated in Appendix E.5., RetroPLEX exhibits only a marginal increase in training time compared to baseline methods.
>
> ---
>
> **[W2]**
>
> We apologize for the lack of a detailed explanation of the model's training process, which may have made it difficult for the reviewers to understand our method. To clarify our approach, we have included a pseudocode for both the NRE retriever and MPC retriever in the attached PDF.
>
> Regarding the NRE retriever, we developed a composition-based formation energy predictor tailored for experimental data. We initially pre-trained the GNN predictor on extensive DFT-calculated data and then fine-tuned it using experimental formation energy data. Using this fine-tuned GNN predictor, we calculate the formation energy of the target material in the dataset and the precursors of all materials in the knowledge base. Subsequently, we calculate the Gibbs free energy and retrieve the top K materials with the most negative Gibbs free energy for each target material. A more detailed training procedure and further analysis of the NRE retriever are provided in Appendix E.2 (page 16).
>
>
> ---
>
> **[Q1]**
>
> The MPC retriever is crucial for effectively leveraging the synthesis practices from the synthesis literature. It identifies reference materials that share similar synthesis recipes with the target material by learning the dependencies among precursors and the correlation between the precursors and the target material. However, despite its effectiveness in identifying reference materials with potentially similar precursor sets, it often misses the thermodynamic relationships between materials. To achieve this, we incorporate the NRE retriever, which identifies reference materials possessing the precursor sets capable of inducing favorable reactions with the target materials by considering thermodynamic forces.
> Utilizing both retrievers, our models can explore a wide range of reference materials that could provide potential recipes for the target materials. We also provide further explanation with qualitative analysis in Q4.
>
> ---
>
> **[Q3]**
>
> Upon reviewer's request, we trained the model with varying dataset sizes: 20%, 40%, 60%, and 80%. As shown in Figure 1 in the attached PDF, as the dataset size increases, we observed a consistent improvement in model performance of Top-10 accuracy. This indicates that the model benefits from having access to more supervision, which likely leads to better generalization and more robust predictions. Moreover, it can be observed that RetroPLEX using only 80% of the training dataset outperforms the graph network using 100% of the training dataset, while our model demonstrates significantly higher performance when both models use 100% of the dataset.
> Additionally, in terms of computational complexity, we tested the performance of the proposed model by reducing the message passing layers (L) of the backbone encoder, Graph Network from 3 to 2 (i.e., RetroPLEX (D:256, L:2)), and by reducing the hidden dimension (D) from 256 to 128 (i.e., RetroPLEX (D:128, L:3)). We observed a decrease in model performance due to the reduced representative power of the model with fewer message passing layers and smaller hidden dimensions. Nevertheless, even with reduced complexity, RetroPLEX (D:256, L:2) and RetroPLEX (D:128, L:3) outperform the Graph Network with 3 message passing layers and a hidden dimension of 256 (i.e., Graph Network (D:256, L:3)). This can be attributed to the information from retrieved materials, which helps the model achieve good results despite its lower complexity. In summary, these two experiments demonstrate that RetroPLEX can effectively perform precursor extraction through trained retrievers, achieving better performance compared to the Graph Network without retrievers. This is possible even with relatively smaller datasets (Figure 1(a)), smaller hidden dimensions (D), and fewer message passing layers (L) (Figure 1(b)).
>
> ---
>
> **[Q4]**
>
> We provide a qualitative analysis of our proposed method for retrosynthesis planning for the target material Na3Dy(PO4)2 in the Table 1 (attached PDF). When only the MPC retriever is used, the method fails to predict the complete precursor set due to insufficient extraction of precursor information from the retrieved materials. However, when both the NRE and MPC retrievers are used, our method successfully predicts the full precursor set for the target material. For example, the NRE retriever allows our method to extract precursor information from NaPO3, which includes a crucial precursor Na2CO3, a direct precursor for the target material. The complementary nature of the retrievers allows our method to effectively extract precursor information from relevant reference materials without explicitly using precursors.

---

> ### Author Response · Authors · 2024-08-11
> **Gentle reminder for author reviewer discussion**
>
> Thank you once again for your valuable review and feedback. We kindly ask if you could please take a moment to confirm whether our rebuttal has adequately addressed your comments and concerns. Thank you for your consideration.

---

> > ### Comment · Reviewer_wQcK · 2024-08-13
> >
> > I acknowledge the response from authors and decide to maintain my score.

---

### Official Review · Reviewer_N6yS · 2024-07-12

**Soundness:** 3
**Presentation:** 3
**Contribution:** 3
**Rating:** 6
**Confidence:** 5

**Summary:**

The authors address the domain of inorganic retrosynthesis by better leveraging existing inorganic retrosynthesis data. They employ attention learning techniques to establish relationships between chemical formulas and precursor formulas. Additionally, they utilize a neural reaction energy predictor to forecast the Gibbs free energy of chemical reactions, thereby refining the candidate list. The integration of these two models significantly enhances the accuracy of the prediction results. This approach has strong applications in the exploration of material synthesis.

**Strengths:**

1. Gibbs free energy is essential for determining if a chemical reaction can occur spontaneously and is more likely to happen. Considering this factor is necessary and has been overlooked in previous works. The inclusion of the neural reaction energy predictor significantly improves prediction results by providing a more accurate assessment of Gibbs free energy for chemical reactions.

2. In the task of inorganic retrosynthesis, predicting the class of a precursor compound from its chemical formula is crucial. There are few models for this task starting from chemical formulas. The authors provide a detailed comparison with models like Roost and CrabNet and evaluate variables such as the number of precursors used. The experiments are thorough and well-documented.

**Weaknesses:**

1. Chemical formula representation issues, detailed in Question 4.
2. Limited improvement over random selection, detailed in Question 5.

**Questions:**

1. The Gibbs free energy data used by the authors is used to train a prediction model, and the authors fine-tuned the model to align with experimental values. However, Gibbs free energy depends on specific conditions such as temperature and pressure during experiments or calculations. It's unclear what the specific conditions of reactions were during this transfer process.

2. In section 3.1, the authors mentioned "The overall training procedure of MPC retriever is in Figure 1 (a)." It seems this should actually refer to Figure 2.

3. In Table 8, under Qualitative Analysis, the authors predict the precursors for Na3Dy(PO4)2 using MPC to find a similar compound Na3Y(PO3)4. However, this compound cannot be found in online materials databases. Moreover, the claimed corresponding precursor sets do not seem capable of synthesizing Na3Y(PO3)4. Please provide the data source and specific synthesis literature for this compound.

4. The authors use chemical formulas as input to their model. However, in practical synthesis tasks, different conditions can yield different structures with the same chemical formula. The authors should clarify how they addressed the issue of duplicate chemical formulas during the data preprocessing stage.

5. In section 4.3, the authors note that "reference materials are randomly retrieved without using trained retrievers" and still achieve a top-1 accuracy of 58%. This high accuracy is surprising and raises questions about the effectiveness of the trained retrievers. The authors should explain why random selection of reference materials performs nearly as well as the trained retrievers and discuss the underlying reasons for this result.

**Limitations:**

The authors use Gibbs free energy to evaluate the synthesis of reference materials. However, predicting Gibbs free energy can be challenging, especially when only the chemical formula is available without specific structures, synthesis conditions, or information about gas release during synthesis. This limitation suggests an area for improvement in making energy predictions more accurate.

Additionally, the method of synthesizing inorganic materials—whether through heating and calcination, microwave, or other techniques—can also affect the choice of precursor compounds. Incorporating these synthesis methods and their specific impacts on precursor selection would further enhance the model's accuracy and applicability. We hope the authors can consider these factors in future improvements to their approach.

---

> ### Author Rebuttal · Authors · 2024-08-07
>
> Thank you for your valuable comments on our work and for recognizing our efforts to address inorganic retrosynthesis using thermodynamic factors! We are more than willing to address each of the specific weaknesses and questions in detail.
>
> ---
>
> **[W1&Q4]**
>
> As the reviewer pointed out, different conditions can yield different structures with the same chemical formula, resulting in duplicate chemical formulas. To consider materials with different structures but the same chemical formula, structural information is essential; however, the dataset we used [1], which was extracted from various publications containing synthesis recipes, lacks this information. This is because many material synthesis experiments are conducted referencing the synthesis recipes of past experiments, and in real-world scenarios, it is common to know only the chemical formula without knowing the structure. In the absence of structural information, we abstracted materials with different structures but the same chemical formula into a single reaction data for our model, so the concern about duplicate chemical formulas, as mentioned by the reviewer, does not exist.
> Additionally, besides having different structures, materials can also have multiple precursor sets depending on the synthesis conditions. We selected the most frequently occurring synthesis routes among those with varying conditions, allowing the model to learn the most likely precursor set for the input target material.
>
> [1] Kononova, Olga, et al. "Text-mined dataset of inorganic materials synthesis recipes." Scientific data 6.1 (2019): 203.
>
> ---
>
>
> **[W2&Q5]**
>
> We fully agree with the reviewer's comment that there needs to be an explanation and analysis for why the performance of the model with random retrievers has nearly the same high accuracy as the model with trained retrievers as shown in Table 3, and for the effectiveness of trained retrievers. We provide an analysis below.
>
> Recall that RetroPLEX employs Graph Network as the backbone encoder. Table 2 in the attached PDF shows that the Graph Network without a retriever (i.e., Graph Network(no retriever)) and the Graph Network with a random retriever (i.e., Graph Network(random)) perform similarly. This indicates that performance Graph Network(random) is mainly due to the Graph Network backbone itself, but not due to the random retriever. On the other hand, in the case of Graph Network(RetroPLEX), which incorporates our proposed trained retrievers, shows consistent improvement over both Graph Network(no retriever) and Graph Network(random). Such improvements of RetroPLEX are even observed with other backbone encoders as well. This confirms the effectiveness of our framework in implicitly extracting useful synthesis recipe information that aids in predicting the synthesis of the target material by utilizing the trained retrievers.
>
> ---
>
> **[Q1]**
>
> Firstly, we apologize for not clearly specifying the temperature and pressure conditions for calculating the Gibbs free energy data. The formation energy from the Materials Project that we used is calculated at 0 K and 0 atm using DFT. The experimental formation energy data is taken from [1], which reports measurements at 298.15 K and 1 atm.
>
> We fully agree with the reviewer's suggestion that using Gibbs free energy specific to each synthesis condition is essential, as material synthesis occurs under varying temperature and pressure conditions. However, calculating Gibbs free energy for every specific condition is impractical due to the considerable cost and time involved. Given that the appeal of AI in material science lies in its efficiency, we have opted to use an abstraction of the complex conditions.
>
> On the other hand, even though the Gibbs free energy we used may not be the exact value for each synthesis condition, we believe it still reflects the trends in chemical reactions since both the formation energy data from the Materials Project and the experimental formation energy data are calculated under consistent temperature and pressure conditions. Nevertheless, we acknowledge the limitations of our proposed method and greatly appreciate the reviewer's valuable comment. We will explore efficient ways to calculate Gibbs free energy considering synthesis conditions and incorporate them into our modeling as part of future work.
>
> [1] Jha, Dipendra, et al. "Enhancing materials property prediction by leveraging computational and experimental data using deep transfer learning." Nature communications 10.1 (2019): 5316.
>
> ---
>
> **[Q2]**
>
> We apologize for the confusion caused by the typo. As the reviewer correctly pointed out, it should refer to Figure 2(a). We appreciate your attention to detail and will make the necessary corrections promptly.
>
> ---
>
> **[Q3]**
>
> First and foremost, we sincerely apologize for any confusion caused by our qualitative analysis. As the reviewer correctly pointed out, upon reviewing Table 8, we found that the correct similar compound retrieved for synthesizing Na3Dy(PO4)2 is NaY(PO3)4, not Na3Y(PO3)4. This critical error of writing Na3 instead of Na led to a misunderstanding. We regret this mistake and any resulting confusion. We have inserted the corrected table (Table 1) into the attached PDF.

---

> ### Author Response · Authors · 2024-08-11
> **Gentle reminder for author reviewer discussion**
>
> Thank you once again for your valuable review and feedback. We kindly ask if you could please take a moment to confirm whether our rebuttal has adequately addressed your comments and concerns. Thank you for your consideration.

---

> > ### Comment · Reviewer_N6yS · 2024-08-12
> >
> > Thank you for your thoughtful rebuttal and for addressing the concerns raised in the initial review. I appreciate the effort you put into clarifying the issues.
> >
> > Regarding the discussion on the precursor compound in the Qualitative Analysis section, you mentioned that the actual example compound is NaY(PO3)4. However, after researching this compound, Na2CO3 is indeed one of the raw materials (precursors) used to synthesize NaY(PO3)4, according to the synthesis method mentioned in the literature [1]. This differs from the precursors you provided in the paper, leading to less effectiveness of NRE retriever. Although this mistake may have originated from the comparison article’s dataset, it is crucial to note that factual errors should not be present as examples in the paper. If the issue indeed stems from the dataset, I recommend considering the use of a different case study to avoid this discrepancy.
> >
> > Your explanations have successfully addressed most of my other concerns, and I hope these comments will be helpful in improving your manuscript.
> >
> > [1] M. El Masloumi, et al. Structure and luminescence properties of silver-doped NaY(PO3)4 crystal. Journal of Solid State Chemistry, 11(181), 2008.

---

> > > ### Author Response · Authors · 2024-08-13
> > >
> > > Thank you so much for your valuable feedback. I'm delighted to hear that most of the concerns, including the weaknesses you highlighted, have been addressed.
> > >
> > > Regarding the missed precursor for synthesizing NaY(PO₃)₄, after realizing that Na₂CO₃ is indeed a necessary precursor for this synthesis, we reviewed the code and dataset preprocessing procedures. We discovered that during the extraction of precursor information from paper [1], Na₂CO₃ was omitted due to an issue within the dataset itself, as you speculated, which led to the factual error. As you pointed out, such errors should not appear in the manuscript, and thanks to your thorough review, we were able to identify and correct this issue. Moving forward, we will take extra care in preparing the manuscript.
> > >
> > > Additionally, we are providing another qualitative analysis. The table below shows the results of our model using both MPC and NRE retrievers for $Pb_9[Li_2(P_2O_7)_
> > > 2(P_4O_{13})_2]$.
> > >
> > > We were able to retrieve $Li_{2}CO_{3}$ and  $NH_{4}H_{2}PO_{4}$ through the MPC retriever, and $NH_{4}H_{2}PO_{4}$ and $PbO$ were identified by using the NRE retriever, which retrieved $Pb_{3}(PO_{4})_{2}$. This demonstrates how the NRE retriever and MPC complement each other to enhance prediction accuracy.
> > >
> > > Moreover, we have included the DOI for each material, sourced directly from the raw data, below the table.
> > >
> > > Thank you once again for your valuable feedback. I hope that this addresses your concerns.
> > >
> > >
> > >
> > >
> > > | **Model**         | **Retriever** | **Retrieved Material** | **Corresponding Precursor Sets**     | **Predicted Precursor Set(Output)**               |
> > > |-------------------|---------------|------------------------|--------------------------------------|-------------------------------------------|
> > > | MPC + NRE         | MPC           | $LiNaPbPO$         | {$Li_{2}CO_{3}$, $H_{3}PO_{4}$,$Na_{2}CO_{3}$, $Pb_{3}O_{4}$ }  | {$NH_{4}H_{2}PO_{4},Li_{2}CO_{3}, PbO $} |
> > > | (RetroPLEX)       | MPC              | $Li_{0.5}Na_{0.5}PO_3$ |{$Li_{2}CO_{3},NH_{4}H_{2}PO_{4}, NaPO_{3}$}|
> > > |                   | MPC              |$Li_{3}V_{1.92}Al_{0.08}(PO_{4})_{3}$ | {$Al, V_{2}O_{5}, LiH_{2}PO_{4}$}      |
> > > |-|-|-|-|-|
> > > |                   | NRE           | $Pb_{3}(PO_{4})_{2}$     | \{$PbO, NH_{4}H_{2}PO_{4}$ } |                                           |            |
> > > |                   |NRE               | $Li_{3}P$               | \{$P, Li$\}                     |                                           |
> > > |                   |NRE               | $PbP_{7}$                  | \{$P, Pb$\}                            |                                           |
> > >
> > >
> > > ---
> > >
> > >
> > >
> > > $Pb_9[Li_2(P_2O_7)_
> > > 2(P_4O_{13})_2]$  : 10.1039/c7dt00509a
> > >
> > > $LiNaPbPO$ : 10.1016/s0022-3093(03)00171-6
> > >
> > > $Li_{0.5}Na_{0.5}PO_3$ : 10.1016/s0022-3093(01)00655-x
> > >
> > > $Li_{3}V_{1.92}Al_{0.08}(PO_{4})_{3}$ : 10.1016/j.electacta.2010.12.063
> > >
> > > $Pb_{3}(PO_{4})_{2}$ : 10.1103/physrevb.73.024429
> > >
> > > $Li_{3}P$ : 10.1021/cm0513379
> > >
> > > $PbP_{7}$  : 10.1039/c4dt01539h
> > >
> > > [1] M. El Masloumi, et al. Structure and luminescence properties of silver-doped NaY(PO3)4 crystal. Journal of Solid State Chemistry, 11(181), 2008.

---

> > > > ### Comment · Reviewer_N6yS · 2024-08-13
> > > >
> > > > Thank you very much to the authors for the timely and thorough response. The new examples indeed demonstrate the role of the NRE retriever effectively. All my concerns have been addressed, and I will adjust my score to 6.

---

> > > > > ### Author Response · Authors · 2024-08-13
> > > > >
> > > > > We appreciate the reviewer recognizing our efforts and deciding to raise the score. Thank you also for your prompt response to our rebuttal. We are sincerely grateful for this!

---

### Official Review · Reviewer_moYU · 2024-07-13

**Soundness:** 3
**Presentation:** 3
**Contribution:** 3
**Rating:** 7
**Confidence:** 5

**Summary:**

The paper introduces RetroPLEX, a method for inorganic retrosynthesis planning. It extracts precursor information from retrieved reference materials implicitly. The authors use attention layers to extract information from the reference material and design a neural reaction energy (NRE) retriever to provide complementary reference materials. Extensive experiments demonstrate the effectiveness of implicit extraction of precursor information and NRE retriever in discovering novel synthesis recipes.

**Strengths:**

Originality: RetroPLEX presents a new approach to inorganic retrosynthesis planning by implicitly extracting precursor information from retrieved reference materials. This deviates from traditional methods that rely on explicit utilization of precursor information.

Quality: The authors provide a comprehensive evaluation of RetroPLEX, including assessments in realistic scenarios, which demonstrates its effectiveness in discovering novel synthesis recipes.

Clarity: The paper is well-written and easy to follow, with clear explanations of the methodology and results.

Significance: The proposed method has significant implications for material science, as it can aid in the discovery of new materials and their synthesis routes.

**Weaknesses:**

No specific complaint. The work is highly praised for its comprehensiveness, thoroughness, and originality. It is supported by ample evidence that attests to its quality. The following section will include questions related to the work.

**Questions:**

How does RetroPLEX handle situations where there are multiple possible synthesis routes for a target material?

How do the authors plan to address the potential issue of overfitting, given the complexity of the model?

**Limitations:**

The authors acknowledge limitations of their work, such as the importance of incorporating precursor information from a broader range of reference materials.

---

> ### Author Rebuttal · Authors · 2024-08-07
>
> Thank you for your high praise and acknowledgment of the novelty of our work in inorganic retrosynthesis. We are more than willing to address your two questions in detail.
>
> ---
> **[Q1]**
>
> As the reviewer noted, actual material synthesis processes often involve multiple possible synthesis routes for a target material based on synthesis conditions, making it crucial to address such scenarios. However, considering the increased complexity involved in synthesis conditions, we opted to follow previous work [1] and focus on a single synthetic route for each target material and precursor set. Specifically, each target material is associated with multiple synthesis routes, gathered from various literature sources, each with distinct experimental conditions. Among the multiple synthesis routes, we selected the most frequently occurring synthesis routes among those with varying conditions, allowing the model to learn the most likely precursor set for the input target material.
>
> On the other hand, recognizing the importance of synthesis conditions, we are planning future work to incorporate these conditions into our modeling. To achieve this, we intend to develop a separate model to predict the sequence of synthesis steps and corresponding conditions from an inorganic synthesis dataset. This approach will enable us to handle unseen materials without synthesis condition information by learning both the target material and synthesis conditions jointly, allowing the model to manage multiple possible synthesis routes.
>
> [1] He, Tanjin, et al. "Precursor recommendation for inorganic synthesis by machine learning materials similarity from scientific literature." Science advances 9.23 (2023): eadg8180.
>
> ---
>
> **[Q2]**
>
> RetroPLEX initially trains two retrievers (i.e., the MPC retriever and the NRE retriever) in advance to simplify the complexity of the model by leveraging the materials retrieved from these trained retrievers. However, as the reviewer pointed out, the model may still face potential overfitting issues. To address this, we applied traditional techniques in machine learning such as L2 regularization and early stopping during the training process.

---

> ### Author Response · Authors · 2024-08-11
> **Gentle reminder for author reviewer discussion**
>
> Thank you once again for your valuable review and feedback. We kindly ask if you could please take a moment to confirm whether our rebuttal has adequately addressed your comments and concerns. Thank you for your consideration.

---

> > ### Comment · Reviewer_moYU · 2024-08-12
> >
> > Thank you for your rebuttal which addresses the questions. I remain my initial assessment of RetroPLEX as a technically sound paper with significant implications for material science, and I recommend its acceptance for publication.

---

> > > ### Author Response · Authors · 2024-08-12
> > >
> > > We are delighted that our work has been recognized, and we sincerely appreciate your compliments and encouragement. Thank you!

---

### Author Rebuttal · Authors · 2024-08-07

Dear reviewers, thank you for your valuable comments on our work. We are more than willing to address each of the weaknesses and questions in detail. Additionally, we have attached a PDF file that includes a qualitative analysis, experiments, and pseudocode for the rebuttal.

---

### Comment · Area_Chair_3EwG · 2024-08-13

Dear reviewers,

As the deadline for the discussion period approaches, could you please take a moment to confirm whether the author’s rebuttal has addressed your concerns, or at least indicate that you have reviewed it?

Best,
AC

---

### Decision · Program_Chairs · 2024-09-25

**Decision:**

Accept (poster)

**Comment:**

This paper introduces RetroPLEX, an inorganic retrosynthesis planning method that implicitly extracts precursor information from retrieved reference materials. The authors utilize attention layers to extract relevant information and design a neural reaction energy predictor to estimate the Gibbs free energy of chemical reactions. Extensive experiments demonstrate the effectiveness of implicit precursor extraction and the NRE predictor in discovering novel synthesis routes.

The reviewers found this work universally interesting and timely, given the growing capabilities in exploring material synthesis. Despite some minor concerns, this is a strong paper that I recommend for acceptance to NeurIPS.